# Non-Typeable *Haemophilus influenzae* Invade Choroid Plexus Epithelial Cells in a Polar Fashion

**DOI:** 10.3390/ijms21165739

**Published:** 2020-08-10

**Authors:** Christian Wegele, Carolin Stump-Guthier, Selina Moroniak, Christel Weiss, Manfred Rohde, Hiroshi Ishikawa, Horst Schroten, Christian Schwerk, Michael Karremann, Julia Borkowski

**Affiliations:** 1Pediatric Infectious Diseases, Department of Pediatrics, Medical Faculty Mannheim, Heidelberg University, Theodor-Kutzer-Ufer 1-3, D-68167 Mannheim, Germany; christian.wegele@medma.uni-heidelberg.de (C.W.); Carolin.stump-guthier@medma.uni-heidelberg.de (C.S.-G.); selina.moroniak@medma.uni-heidelberg.de (S.M.); horst.schroten@medma.uni-heidelberg.de (H.S.); christian.schwerk@medma.uni-heidelberg.de (C.S.); michael.karremann@medma.uni-heidelberg.de (M.K.); 2Department of Medical Statistics and Biomathematics, Medical Faculty Mannheim, Heidelberg University, Theodor-Kutzer-Ufer 1-3, D-68167 Mannheim, Germany; christel.weiss@medma.uni-heidelberg.de; 3Central Facility for Microscopy, Helmholtz Centre for Infection Research, Inhoffenstraße 7, D-38124 Braunschweig, Germany; manfred.rohde@helmholtz-hzi.de; 4Laboratory of Clinical Regenerative Medicine, Department of Neurosurgery, Faculty of Medicine, University of Tsukuba, 1-1-1Tennodai, Tsukuba, Ibaraki 305-8575, Japan; ishi-hiro.crm@md.tsukuba.ac.jp

**Keywords:** choroid plexus, blood–cerebrospinal fluid barrier, NTHI, *Haemophilus influenzae*, host pathogen interaction, meningitis

## Abstract

Non-typeable *Haemophilus influenzae* (NTHI) is a pathogen of the human respiratory tract causing the majority of invasive *H. influenzae* infections. Severe invasive infections such as septicemia and meningitis occur rarely, but the lack of a protecting vaccine and the increasing antibiotic resistance of NTHI impede treatment and emphasize its relevance as a potential meningitis causing pathogen. Meningitis results from pathogens crossing blood–brain barriers and invading the immune privileged central nervous system (CNS). In this study, we addressed the potential of NTHI to enter the brain by invading cells of the choroid plexus (CP) prior to meningeal inflammation to enlighten NTHI pathophysiological mechanisms. A cell culture model of human CP epithelial cells, which form the blood–cerebrospinal fluid barrier (BCSFB) in vivo, was used to analyze adhesion and invasion by immunofluorescence and electron microscopy. NTHI invade CP cells in vitro in a polar fashion from the blood-facing side. Furthermore, NTHI invasion rates are increased compared to encapsulated HiB and HiF strains. Fimbriae occurrence attenuated adhesion and invasion. Thus, our findings underline the role of the BCSFB as a potential entry port for NTHI into the brain and provide strong evidence for a function of the CP during NTHI invasion into the CNS during the course of meningitis.

## 1. Introduction

*Haemophilus influenzae* (*H. influenzae*) is a human-restricted Gram-negative bacterium. Due to the composition of the polysaccharide capsule *H. influenzae* can be classified into six capsular serotypes from A to F. The polysaccharide capsule is a major virulence factor of *H. influenzae* [1] and protects against opsonization by neutrophils, providing encapsulated strains an advantage in survival over non-encapsulated strains [2]. Encapsulated variants can become capsule-deficient due to phase variation that may switch off capsule expression [3]. Some strains contain a complete or partial locus for capsule production, but lack a functional *bexA* gene, encoding a protein responsible for the capsular polysaccharide export to the cell surface [4]. By conventional means, those strains are also non-serotypeable and appear to be non-typeable *H. influenzae* (NTHI) [5]. True NTHI, however, completely lack the genomic capsule locus [6]. 

*H. influenzae* can cause severe invasive diseases including meningitis, pneumoniae and sepsis. Prior to the recommendation of the *H. influenzae* type B (HiB) vaccine in 1990, HiB was the major cause of bacterial meningitis in young children. Invasive HiB infections remain critical in countries with low vaccination rates, but even in high-income countries HiB is still prevalent [7,8,9]. However, type B invasive infections decreased >90% in countries with routine HiB immunization in the national immunization program [10,11]. Since the capsule B targeted vaccine has no impact on other serotypes, today, the most common encapsulated *Haemophilus* causing invasive infections is serotype F (HiF) [8,12]. A predominant incidence of *Haemophilus* infections, however, is caused by NTHI strains [5,12,13,14,15,16].

NTHI commensally colonize the respiratory tract and occasionally become pathogenic causing localized diseases. In contrast to HiB infections, invasive NTHI infections tend to affect susceptible populations rather than healthy individuals, such as persons with impaired immune function (premature infants, elderly adults, patients with cancer or under immunosuppressive therapy) and with chronic cardiovascular diseases, respiratory insufficiencies and metabolic diseases. Thus, in high income countries, increased life expectancy may indirectly augment the population of at-risk individuals due to a higher proportion of elderly, and NTHI are considered pathogenic rather than commensal [5,17]. In children, NTHI are the most common cause of otitis media, and, in adults, the majority of chronic obstructive pulmonary disease (COPD) is exacerbated by involvement of NTHI [18]. Moreover, NTHI can cause sinusitis, epiglottitis, bronchitis, pneumonia and less frequently bacteremia and meningitis. Even though meningitis and septicemia occur rarely in children with predisposing conditions such as head trauma, cerebrospinal fluid (CSF) leakage and immune compromise, or in neonates due to NTHI colonization of the maternal genital tract. More rarely, in healthy children [19], several publications demonstrate that NTHI is a potential pathogen able to cause meningitis in children [19,20,21,22,23], a life-threatening disease difficult in treatment due to increasing prevalence of NTHI antibiotic resistance [24]. Additionally, the German National Reference Center for Meningococci and *H. influenzae* provides data from 2018 revealing a frequency of 63% of CSF infections (17 of 27 CSF samples) to be NTHI caused infections indicating the virulence of NTHI (http://www.meningococcus.uni-wuerzburg.de/startseite/berichte/berichte-h-influenzae/daten-2018/). 

Since the capsule is the major virulence factor of *H. influenzae,* NTHI evidently own features other than a polysaccharide capsule, which equip them to survive in human serum to cause meningitis or other invasive infections. Encapsulated *Haemophilus* and NTHI share several virulence factors, which, however, seem to be more relevant in NTHI [25,26,27,28,29,30,31,32,33,34,35]. Adhesins play an important role during the course of pathogenesis. They can be divers in structure, such as trimeric autotransporters (Hmw1, Hmw2, Hia and Hap), surface protrusions (T4SS and Hif), surface proteins (outer membrane proteins (Omp) P1, Omp P2, Omp P4, Omp P5 and Omp P6 and proteins E and F) and the lipooligosaccharide (reviewed in [36,37]). In NTHI, the latter can contribute to compensate the function of the lacking capsule by surface decoration mediating complement resistance [30]. LpsA, a glucosyltransferase responsible for addition of a hexose (glucose or galactose) to the distal heptose (linkage to the heptose either in a β1–2 or β1–3 fashion) of the inner core of the lipopolysaccharide molecule [38], has been reported to play a crucial role in serum resistance and the internalization in epithelial cells [30]. 

Prerequisite for disease development is colonization, epithelial adherence, evasion of the immune system and finally penetration of cellular barriers. Several studies examined NTHI respiratory colonization, persistence and invasion, and they identified signaling and trafficking pathways involved in NTHI pathogenesis [39]. As meningitis is a less frequently observed clinical manifestation of NTHI invasive disease, less research has been done in this field. Experimental HiB-meningitis in infant rhesus monkeys, however, identified the CP as the primary site of invasion into the CSF and the site of exhibiting the earliest histopathological lesions. Thus, *Haemophilus* probably enters the CSF from inflamed plexal capillaries resulting in plexitis and ventriculitis [40,41,42]. In vitro data also indicate that *H. influenzae* invades the CSF via the BCSFB [43]. Several other meningitis causing bacteria have been reported to enter the brain via the CP in vivo [44,45,46,47,48,49,50,51,52,53,54,55] and in vitro [43,56,57,58,59], allowing the conclusion that penetration of the BCSFB constituted by the epithelial lining of the CP epithelial cells from the blood to the brain is a pathophysiological step from bacteremia to meningitis [56,57]. To the best of our knowledge, no data are available clearly identifying the site of invasion for NTHI during the progress of meningitis and no other study so far addressed the mechanism for NTHI entering the central nervous system (CNS) via the BCSFB. NTHI caused meningitis, even if a rare event, is a severe life-threatening disease difficult in treatment because of the antibiotic resistant prevalence of NTHI [24,60]. Enlightening the mechanisms of how NTHI enter the CNS prior to meningeal inflammation may provide opportunities for treatment. 

In the present study, a previously developed in vitro model of the BCSFB based on a human CP papilloma (HIBCPP) cell line representing well the barrier characteristics of CP epithelial cells, such as polarity, little permeability for macromolecules and a strong transepithelial electrical resistance (TEER), was used to analyze NTHI invasion [56,61,62]. NTHI adhered to and polarly invaded into cells of the human BCSFB in vitro (HIBCPP) as the bacteria enter the cells almost exclusively from the basolateral blood-facing side. In contrast to NTHI, encapsulated HiB and HiF reference strains displayed moderate invasion rates. Fimbriae clearly attenuated invasion and adhesion rates, indicating an important role during NTHI invasion. Our findings clearly elucidate the relevance of the CP for the entry of NTHI into the CNS during meningitis pathogenesis.

## 2. Results

### 2.1. Characterization of Different Haemophilus Strains for Capsule Markers and Virulence Factors

Despite the absence of the capsule, NTHI are able to withstand the human immune system and cause meningitis in vivo. Presently, NTHI are the most common pathogen causing CSF infections of all *Haemophilus* types in Germany (data 2018 published by the NRZMHi, Würzburg, Germany; http://www.meningococcus.uni-wuerzburg.de/startseite/berichte/berichte-h-influenzae/daten-2018). Therefore, in the present study, we addressed the ability of NTHI to invade the CSF via the BCSFB in vitro. For comparison, invasion rates of HiF, which is the “second” most common serotype, and HiB were analyzed in parallel. 

NTHI 3144, HiB 1203 and HiF 825 were originally isolated from the CSF of patients suffering from meningitis, whereas NTHI strains 45 p+ and 45 p− are nasopharyngeal isolates. To verify the authenticity of the strains, we performed PCR on genomic DNA. Strains were tested for the presence of genes located in the capsule locus, including *bexA,* which is responsible for the export of capsular components, as well as the genes for capsular typing, *capB* and *capF* (Figure 1A). 

As supposed, the NTHI strains contained neither the *bexA* gene nor the genes for the capsular serotype B or F. Results for strains HiB 1203 and HiF 825 were in accordance with their predicted serotype status and showed the supposed PCR product for *bexA* amplification.

Fimbriae (precisely, the so-called LKP pili) mediate agglutination of human erythrocytes by binding to the Anton blood group antigen [63,64]. Presence of fimbriae was analyzed on genomic level by PCR using primers against the major pilus structural subunit, *hifA*, but also by hemagglutination of freshly isolated human erythrocytes indicating actual fimbria expression. The phase variants 45 p+ and 45 p− have been isolated from the parental strain AAr45 by selection of hemagglutination with the erythrocyte selection technique. In 45 p− a phase variation in the *hifA* gene led to loss of the fimbriae. The effect of this phase variation could be confirmed by PCR (Figure 1A) and hemagglutination (Appendix A). NTHI strain 3144 and the HiB 1203 isolate showed neither hemagglutination (Appendix A) nor a PCR product with *hifA* primers (Figure 1A). No hemagglutination was also detected for the HiF 825 isolate (Appendix A), even though it contained the *hifA* gene sequence (Figure 1A).

Next, the genomic presence of several described virulence factors was analyzed by PCR on genomic DNA (Figure 1B). Genes encoding for Omp P1 (*omp1*), Omp P2 (*omp2*), Omp P4 (*hel*), Omp P5 (*ompA*) and Omp P6 (*omp6*) occurred in all five isolates. The same was true for the genomic sequence encoding for the LPS glucosyltransferase LpsA. The Hia gene sequence could only be detected in the NTHI strain 3144 and the HiF 825 isolate. In strain HiB 1203, PCR resulted in an amplification product of ~500 bp instead of the supposed 766 bp. Since Hia has been reported not to be present in the HiB serotype [32,65], the truncated amplificate probably was due to unspecific amplification due to a high similarity (81%) and identity (72%) of Hia to the *H. influenzae* type B surface fibril (Hsf) [66] The forward oligonucleotide primer also binds to the Hsf genomic sequence of at least two published sequences of type B strains (accession number U41852, CBW30091). 

### 2.2. Maintenance of Barrier Integrity upon HIBCPP Exposure to NTHI

To enter the immune privileged site of the CSF, pathogens have to cross the BCSFB from the blood, a pathophysiological step from bacteremia to meningitis [43]. A previously established human BCSFB model consisting of a human CP epithelial cell line (HIBCPP cells) was used to address invasion of NTHI into the CSF. HIBCPP cells combine a high TEER and a low permeability for macromolecules such as inulin [56], representing the tightly sealed part of the BCFSB at the CP. In the inverted cell culture model, bacteria can invade the cells from the basolateral side, which in vivo is adjacent to blood vessels and simulates the physiological path of bacteria into the brain via the BCSFB to cause meningitis. The standard culture model resembles the situation, when bacteria are already present in the CSF and get in contact with the apical (brain facing) side, both HIBCPP model variants have been described in detail elsewhere [56,67]. 

To verify that invasion would not affect the barrier integrity of HIBCPP cells during the course of infection, TEER and permeability of the HIBCPP cell layer were evaluated. TEER values were determined at the beginning of the experiment and after 6 h of infection. As Figure 2 confirms, the TEER of HIBCPP cells did not decrease after 6 h of infection in contrast to uninfected cells in the inverted (Figure 2A) and the standard (Figure 2B) cell culture model. Additionally, barrier integrity is indicated by paracellular inulin flux and was not influenced by infection with *Heamophilus* strains for 6 h when compared to the control (Figure 2C,D). Whereas a very low paracellular flux was observed in the inverted model during the 6 h of experiments (Figure 2C), standard culture inserts exhibited a higher inulin flux rate (Figure 2D), indicating some leakage during the course of experiments. Upon treatment of HIBCPP with Cytchalasin D, TEER drops dramatically and permeability massively increases, demonstrating barrier impairment (Appendix A). Vitality of the cells at the end of infection was not affected, as verified with a live/dead assay yielding no differences between infected and uninfected cells (Appendix A).

### 2.3. NTHI Polarly Invade into Human CP Epithelial Cells

Next, the invasion of NTHI into cells of the BCSFB was addressed. Bacterial invasion was detected by double immunofluorescence (DIF) staining, scanning electron microscopy (SEM) and transmission electron microscopy (TEM).

Analyzing inserts of the inverted cell culture filter model, NTHI were able to enter HIBCPP cells from the basolateral side, and single bacteria even transmigrated through the cell layer followed by adhesion to the apical cell side, which in vivo would imply their entry into the CSF (Figure 3A–C). Bacteria were found distributed in single scattered specimen predominantly invaded into the cells. A similar distribution was observed during invasion of the encapsulated HiB 1203 and HiF 825 CSF isolates into HIBCPP cells (Figure 3D,E) [43]. Figure 4 displays representative areas of the cell culture inserts infected with NTHI and strains HiB 1203 and HiF 825 analyzed by TEM and verifies invasion events from the physiological relevant basolateral side. Using TEM, successfully completed basolateral invasion events could be documented, and the different strains could be regularly observed within the cells of the HIBCPP barrier (Figure 4A–F). Figure 4B represents bacteria located in different phases. One bacterium adheres to the basolateral side of the cell, another one is just invading and a third is already located inside of the cell.

Previously, a characteristic polar invasion into HIBCPP cells preferentially from the basolateral side has been shown for several meningitis causing bacteria including *Escherichia coli* [58], *Listeria monocytogenes* [57], *Neisseria meningitidis* and *Streptococcus suis* [56]. Furthermore, Häuser et al. demonstrated a similar polar invasion for *H. influenzae* 770235 wild type strain and corresponding acapsular and non-fimbriated variants [43]. 

To discover whether NTHI also invade preferentially from the basolateral cell side following the pathophysiological way from the blood into the brain, we analyzed invasion of NTHI into HIBCPP cells from the apical side in the standard cell culture model. 

When applied to the apical side of HIBCPP cells, NTHI were found almost only extracellularly adhering to the cells (Figure 5A–C). Invasion events upon infection from the apical side were observed very rarely in the standard culture infected cells. This was also observed for the encapsulated HiB 1203 and HiF 825 CSF isolates (Figure 5D,E). Consequently, NTHI as well as the analyzed HiB and HiF isolates entered HIBCPP cells in the predicted polar fashion (Figure 5A–E). Additional SEM picture sections clearly show NTHI strains 45 p+, 45 p− and 3144 located at the microvilli covered apical cell side (Figure 6A–C), which is in accordance with the extracellular adhered bacteria detected in the DIF. TEM was performed not to miss possible invasion events. In several pictures, we were able to find bacteria adhering to the cells sitting on or next to microvilli of the HIBCPP cells, but not invaded into the cells (Figure 6D–H). Only extremely rarely bacteria were found located between the cells, probably where barrier integrity was not fully maintained (data not shown). Thus, TEM further supports a polar invasion of NTHI into HIBCPP cells from the basolateral side. The same was true for HiB 1203 and HiF 825 CSF isolates, for which invasion from the basolateral side has previously been shown [43]. Both strains hardly show invasion when applied to the apical side (Figure 6G,H). Noteworthy, although in the inverted cell culture model the access of bacteria to the basolateral cell side is limited to areas at the membrane pores (~14% compared to the standard culture), invasion was almost exclusively observed from the basolateral cell side. 

Next, we quantitatively analyzed the invasion events in both the inverted and the standard culture models of HIBCPP, as recently described [67]. All strains show significantly higher invasion rates (around 30-800-fold) when bacteria were applied to the basolateral (blood-facing) cell side (Figure 7A,B). Thus, we quantitatively confirmed polar invasion for NTHI into cells of the BCSFB. For the encapsulated HiB and HiF strains, polar invasion into HIBCPP cells could also be documented (Figure 7B). 

### 2.4. Adhesion of NTHI and H. Influenzae to HIBCPP Cells

Figure 5 and Figure 6 demonstrate that NTHI adhered to HIBCPP cells, when infection was conducted from the apical (brain-facing) side. We also quantified adhesion events and calculated corresponding adhesion rates (Figure 7C). Of note, adhesion to HIBCPP cells can only be reliably analyzed following apical infection in the standard cell culture. In the inverted cell culture model, the basolateral side of the cells directly faces the insert membrane; thus, adhesion to the basolateral cell side occurs only at the membrane pores and is rarely observed. The results show that the proportion of adhesion of the individual strains approximately resembles those received for invasions (Figure 7C). 

### 2.5. A Capsule Attenuates Invasion of H. Influenzae into HIBCPP Cells

Several studies have shown that the presence of a bacterial capsule can impair bacterial invasion [43,56,68]. Basolateral invasion rates of the encapsulated strains HiB 1203 and HiF 825 were comparable to recently described data [43]. For the NTHI isolates 3144 and 45 p−, we observed a significantly higher invasion into HIBCPP cells than for the capsule containing isolates (Figure 7A,B), correlating with data by Häuser et al. who showed the *H. influenzae* capsule to apparently reduce invasion in HIBCPP cells [43]. 

### 2.6. Fimbriae Attenuate Invasion and Adhesion into HIBCPP Cells 

Expression of fimbriae by *H. influenzae* can affect invasion [43]. In the present study, the effect of fimbriae could clearly be observed comparing the nasopharyngeal strains 45 p+ and p−, which are genetically identical except for the presence of fimbriae due to phase variation. Basolaterally infected cells showed significantly less invasion for the fimbriated strain 45 p+ in comparison with 45 p−. Absence of fimbriae in the 45 p− variant resulted in invasion rates, which were around three times higher than that of 45 p+ (Figure 7A). For apically infected cells, fimbriae attenuated invasion about 40-fold (Figure 7B) and adherence about 10-fold (Figure 7C). 

## 3. Discussion

As part of the normal flora, NTHI asymptomatically colonizes the mucous membranes of the human respiratory tract. Acquisition among children already is highest during the first year of life [69]. When the bacterium gets access to privileged anatomic sites, the actual commensal may become pathogenic and induce severe infections including otitis media, sinusitis, conjunctivitis, respiratory infections, septicemia and meningitis [70,71]. A critical step during meningitis pathogenesis is the bacterial crossing of brain barriers (BBB or BCSFB) and their entry into the CNS. Several studies investigated the impact of the BBB during *Haemophilus* meningitis [72,73,74,75,76,77]. However, early studies with rhesus monkeys indicate that *Haemophilus* enters the brain via the BSCFB [40,41,42]. Recently, we reported that HiB and HiF CSF isolates invade cells of the human BCSFB upon infection from the physiologically relevant basolateral side [43]. Thus far, we found no data available about the site of CNS entry for NTHI. To our knowledge, this is the first time that cellular mechanisms of CNS invasion for NTHI across the BCSFSB were investigated. In the present study, we demonstrated that NTHI are able to invade human CP epithelial cells in vitro. Due to the design of the used models the results indicate that NTHI as well as HiB and HiF preferentially invade polarly into HIBCPP cells. EM pictures clearly support polar invasion demonstrating the different localizations of bacteria in the standard and inverted HIBCPP culture models. 

Much significance in systemic disease has been attributed to the presence of the capsule of *H. influenzae* since prior to the use of the HiB vaccine encapsulated strains were responsible for the majority of systemic infections [3]. During the invasion process, the capsule is rather hindering and, therefore, often is switched off even in encapsulated individuals [68]. Thereby, adhesins such as Hia present in strain HiF 825 can become uncovered and support adherence, a critical step during disease development. 

Since the implementation of routine immunization against HiB thirty years ago, the picture has changed and NTHI strains have become considerably more relevant in causing bacteremia, meningitis and other forms of invasive disease [78] (http://www.meningococcus.uni-wuerzburg.de/startseite/berichte/berichte-h-influenzae/daten-2018). Several studies show that today NTHI are the most common *Haemophilus* responsible for invasive infections [5,13,14,16]. Thus, virulence factors other than the capsule must be critical in NTHI survival and invasive diseases. Several surface structures have been reported to affect virulence, but no single feature characteristic for all disease-associated strains could be identified. NTHI strains with similar potential for causing disease might possess different combinations of virulence-related genes [79].

Outer membrane proteins have been described to be involved in adherence of encapsulated *H. influenzae* and NTHI. Both Omp P2 (pore forming structure) and Omp P5 (fimbriae forming structure) mediate interaction with human nasopharyngeal mucins [80]. Extracellular matrix proteins (fibronectin, vitronectin and laminin) can be targeted by Omp P4, promoting adhesion and resulting in immune evasion. Omp P5, which is a universally occurring protein in NTHI, provides complement resistance by binding to factor H [31,81,82]. Several Omps have been reported to bind to cellular receptors such as the intercellular adhesion molecule 1 (ICAM1) (Omp P5 [26]), the carcinoembryonic antigen-related cell adhesion molecule 1 (CEACAM1) (Omp P1 [83] and Omp P5 [84]), the laminin receptor at human microvascular endothelial cells of the BBB ((Omp P2) [85]) and Toll like receptor 2 (TLR2) ((Omp P6) [86]). 

All five strains used in this study own the genes for Omp P1, Omp P2, Omp P4, Omp P5 and Omp P6 expression. Thus, the data do not allow any interpretation of how the Omps affect the differences observed in invasion rates. Additionally, thus far, we cannot evaluate how host receptors reported to be targeted by Omps such as CEACAM1 or ICAM1, which are clearly expressed in HIBCPP cells on RNA level [87], play a role during the invasion process at the BCSFB. 

The *Heamophilius* adhesin (Hia) is the major adhesin expressed by a subset of NTHI and only occurs in non-type B *Haemophilus* but not HiB [65]. Hia promotes efficient adherence to a variety of cell types [33,88]. Additionally, a study reported Hia to be present in the majority of NTHI strains that caused meningitis in children although generally occurring rarely in NTHI [20,25]. Thus, the adhesin has been discussed to play a role in the virulence mechanism of meningitis causing NTHI and may promote invasion into cells of the CP. Interestingly, in the present study, two CSF strains (NTHI 3144 and HiF 825) also exhibited the genomic sequence for Hia. Further research may allow interpretation whether the adhesin indeed affects invasion rates at the BCSFB and has an impact on meningitis pathogenesis. The corresponding host receptor for Hia is still unknown, although the structure of the adhesin has been intensively analyzed by the group of St Geme and the identification of the host cell receptor is under way [89,90]. The identity of the receptor may provide opportunities for modulation in treatment.

Fimbriae represent an additional group of virulence factors of *H. influenzae* and play an important role for the early colonization of the respiratory tract of the human being [91]. Stronger adhesion for fimbriated strains has been shown in several studies [92,93,94]. However, a study where HiB isolates from children suffering from invasive infections isolated from systemic sites and the nasopharynx have been matched (identical bacterial characteristics), revealed that invasive HiB had less fimbriae than non-invasive strains [95]. Thus, fimbriae are either not important for HiB invasion or get lost and become dispensable early during the course of infection [91,95]. The latter data are consistent with data from Häuser et al., where a stronger HiB adherence/invasion to/into CP epithelial cells in absence of fimbriae was observed [43]. In the present study, strongest invasion was also observed by NTHI strains (45 p− and 3144) without fimbriae. The strong influence of fimbriae during invasion becomes apparent comparing invasion rates of strains 45 p+ and 45 p−, which are genetically identical besides fimbriae presence [43]. Fimbriation of strain 45 p+ clearly attenuated invasion. The parental strain AAr45 originally was isolated from the nasopharynx [96]. Predisposing conditions including age, viral infections or constant exposure to pollution can contribute to a shift of NTHI from commensal to pathogen [83]. Thus, the strong invasion of the nasopharyngeal strain 45 p− into cells of the BCFSB in vitro should not be considered negligible, since altering conditions such as a parallel viral infection can let the strain become invasive and reach privileged sites, triggering severe disease despite its nasopharyngeal origin. Furthermore, the outcome of an infection may vary in different patients due to individual host factors and to potential antibodies due to a prior contact with other *H. influenzae* strains [30,97]. Thus, infection with strain 45 p− in another patient may lead to severe disease instead of persisting in a commensal nasopharyngeal state [30]. Fimbriae were also found in CSF strain HiF 825. Together with the polysaccharide capsule, both features might contribute to the weak invasion displayed by this strain in vitro. 

Finally, the lack of capsular serotypes in NTHI, together with the absence of fimbriae and the occurrence of the Hia adhesin, as in strain 3144, resulted in increased invasion into HIBCPP cells. Probably other factors support this synergistic effect, since in contrast to encapsulated strains genetic diversity is pronounced among NTHI. Unfortunately, no data regarding the clinical severity caused by the analyzed isolates are available to conclude whether the generated in vitro data are in accordance with the clinical presentation of the patients in vivo. However, NTHI strain 3144 seems to be an interesting candidate to further investigate the penetration of the BCSFB elucidating the pathophysiology of NTHI caused meningitis. 

In summary, the present study could show that NTHI, besides HiF and HiB isolates, can invade cells of the human CP in a polar fashion from the physiological relevant side, supporting the role of NTHI in causing meningitis. The successful invasion and their possible entry into the immune privileged site of the CSF via the BCSFB is affected by several factors including fimbriae, capsule and adhesins. Since genetic diversity is great among NTHI, it is difficult to identify, which features or combinations exactly qualify NTHI to induce severe invasive diseases, and thus needs further investigation. As long as no vaccine against NTHI is available accurate knowledge about the respective isolate on the basis of molecular typing methods and the cellular mechanisms of how NTHI enter anatomical privileged sites will help to understand NTHI pathology and treat infections besides the use of antibiotics. Evidently, this study could elucidate the function of the CP as a relevant entry port of NTHI to cause meningitis.

## 4. Materials and Methods 

### 4.1. Bacterial Strains and Growth Conditions

NTHI strains 45 p+ and 45 p− were kindly provided by Janet R. Gilsdorf. The original strain AAr45 was isolated from the nasopharynx of a patient and is classified as NTHI [96]. 45 p+ and 45 p− are fimbriated and non-fimbriated phase variants, which could be differentiated due to selection of hemagglutination with the erythrocyte selection technique as described [98]. The other three strains (HiB strain H1203, HiF strain H825 and NTHI strain 3144) were kindly provided by the National Reference Center for Meningococci and *Haemophilus influenzae* (NRZMHi, Wuerzburg, Germany). Those strains were originally isolated from the liquor of meningitis patients (CSF isolates). 

All strains were stored at −80 °C in brain heart infusion broth supplemented with 20% glycerol. For infection assays bacteria were grown on chocolate agar with vitox (Oxoid, Wesel, Germany) and cultured at 37 °C and 5% CO_2_ overnight. The following day several colonies were inoculated in Dulbecco’s Modified Eagle Medium (DMEM)/HAM’s F12 1:1 containing 4 mM L-glutamine and 15 mM HEPES (Thermo Fisher Scientific, Darmstadt, Germany) supplemented with 10% fetal calf serum (FCS) (Gibco, Carlsbad, CA, USA) and 5 µg/mL insulin (Sigma-Aldrich, Darmstadt, Germany) (phenol-red free DMEM/F12 10% FCS), washed twice and adjusted to an optical density at 600 nm (OD_600_) containing approximately 1 × 108 colony forming units (CFU) per mL. In parallel with infection assays, bacterial growth after 6 h for each strain was evaluated in phenol-red free DMEM/F12 10% FCS, to which the number of invasion and adhesion events is set in relation to calculate the corresponding invasion and adhesion rates.

### 4.2. Verification of Bacterial Strains

The authenticity of bacteria was confirmed by PCR on genomic DNA and hemagglutination. NTHI are defined as *H. influenzae*, which do not possess the *bexA* gene or a capsule [99]. Therefore, NTHI were tested for the absence/presence of the *bexA* gene, whose expression product is an ATP binding protein, an energy providing component of the capsule export apparatus and, thereby, positive in capsulated and negative in capsule deficient strains [4,100]. Classification of *H. influenzae* serotypes B and F was conducted using appropriate primers against the capsule encoding gene. To identify fimbriated variants, PCR with primers for the *hifA* gene, the major pilin subunit, was performed. For validation the presence of the genes encoding for outer membrane protein (OMP) Omp P2, Omp P4, Omp P5 and Omp P6 was analyzed. Oligonucleotides used for PCR are listed in Table 1. PCR was performed as follows: initial denaturation (95 °C, 2 min) and subsequently 35 cycles of denaturation (96 °C, 30 s), annealing (55 °C, 30 s) and extension (72 °C, 1 min), followed by a final extension step (72 °C, 7 min). Primers for *omp1* and *ompA* were designed using Primer 3 software [101].

**Table 1 ijms-21-05739-t001:** Oligonucleotides used for verification of bacterial strains.

Gene Symbol	Forward Primer	Reverse Primer	Size (bp)	Reference
*bexA*	CGTTTGTATGATGTTGATCCAGACT	TGTCCATGTCTTCAAAATGATG	343	[100]
*capB*	GCTTACGCTTCTATCTCGGTGAA	ACCATGAGAAAGTGTTAGCG	370	[6]
*capF*	GCTACTATCAAGTCCAAATC	CGCAATTATGGAAGAAAGCT	450	[102]
*hifA*	ATGAAAAAAACACT(AT)CTTGGTAGC	TTAT(CT)CGTAAGCAATT(GT)GGAAACC	650	[103]
*lpsA*	TTGAATATCGTTTAGCAC	GCGTGGCGACAATTAGGC	994	[38]
*hia*	CGCGGCTTGGGCTGGGTCATTTCT	TCAGCCGTACCGTCAGCATTCAGTTCA	766	[32]
*omp1*	TGATAAATTCGCGCTGGGTG	GGCAGTGCGGTCAGTAAAAT	454	this study
*omp2*	ATAACAACGAAGGGACTAACG	TCTACACCGAATAATACTGCT	1000	[104]
*hel*	ATTGGATCCGAATTCTTAAAAGGAAT	ATTAAATATTGGATCCAGTAAAAACTGAGC	1047	[105]
*ompA*	CGTGCCTCTGGTTTATTTGC	GCGATTTCTACACGACGGTC	581	forward modified from [81] reverse this study
*omp6*	ACTTTTGGCGGTTACTCTGT	TGTGCCTAATTTACCAGCAT	273	[100]

Abbreviations: *bexA*, capsule expression; *capB*, capsular B antigene; *capF*, capsular F antigene; *hifA*, *H. influenzae* fimbrial gene; *lpsA*, lipopolysaccharide glycosyl transferase; *hia*, *H. influenzae* adhesin; *omp*, outer membrane protein; *hel*, *H. influenzae* outer membrane lipoprotein e(P4); *ompA*, outer membrane porin A.

Hemagglutination due to the presence of hemagglutinating fimbriae was performed using fresh blood samples of healthy donors (permitted by local ethical review committee, 2015-632N-MA). Red blood cells were separated, washed twice with NaCl 0.9% and mixed with bacterial samples resuspended in NaCl 0.9%. Hemagglutination was evaluated after five minutes of incubation.

### 4.3. Standard and Inverted Cell Culture of HIBCPP Cells

Cultivation of HIBCPP cells was performed as previously described [43,56,67]. Briefly, HIBCPP cells were grown in DMEM/HAM’s F12 1:1 containing 4 mM glutamine and supplemented with 5 μg/mL insulin, penicillin (100 U/mL), streptomycin (100 μg/mL) and 10% FCS (HIBCPP-medium). 

HIBCPP cells were seeded on cell culture filter inserts with a pore diameter of 3.0 μm, pore density of 2.0 × 10^6^ pores per cm^2^ and membrane diameter of 0.336 cm^2^ (Greiner Bio-One, Frickenhausen, Germany). For the standard cell culture, cells were seeded in the upper compartment of the filter inserts; for the inverted cell culture, cells were grown on “the bottom” of the cell culture filter inserts flipped over before seeding and put in correct orientation the day after seeding [67]. Cells were supplied with fresh medium every second day. Cells between passage 20 and 32 were used for infection experiments.

### 4.4. TEER Measurements

To determine barrier function, from Day 4 TEER values were measured continuously every day using an epithelial tissue voltometer with the STX-1 electrode system (Millipore, Schwalbach, Germany). Once HIBCPP exhibited TEER values around 70 Ω × cm^2^ HIBCPP-medium supplied with only 1% FCS without antibiotics was applied.

### 4.5. Infection Assays and Barrier Integrity

Infection experiments with subsequent determination of barrier permeability were performed as previously described [43]. Briefly, cell culture inserts exhibiting appropriate TEER values of 300–800 Ω × cm^2^ upon serum starvation overnight, representing a cell layer with approximately 4 × 10^5^ cells per culture insert, were infected with a multiplicity of infection (MOI) of ~20 for 6 h. Bacteria were applied to the upper compartment, which represents an apical infection in standard cell culture system and a basolateral infection in the inverted cell culture system due to the different growth areas on the upper and the lower side of the membrane used for cell cultivation. Barrier integrity was monitored by the measurement of TEER values before and at the end of infection. In addition, barrier integrity was evaluated determining the paracellular permeability. Therefore, the passage of FITC-coupled inulin (100 μg/mL, average molecular weight of 3000–6000; Sigma, Deisenhofen, Germany) from the upper to lower insert compartment was monitored using a Tecan Infinite M200 Multiwell reader (Tecan, Switzerland), as previously described [57]. 

### 4.6. Double Immunofluorescence Staining and Subsequent Determination of Bacterial Invasion

To determine bacterial invasion rates infected, cells were double immunofluorescence (DIF) stained, as previously described [56] with small modifications. After 6 h of infection, extracellular bacteria were detected with the polyclonal rabbit-anti-HiB described in [43] for HiB and HiF strains and with the polyclonal rabbit-anti-NTHI (Immunoglobe Antikörpertechnik GmbH, Himmelstadt, Germany) for NTHI. Following the staining of extracellular bacteria with Alexa fluor anti-rabbit 594 (1:250, Thermo Fisher, Rockford, IL, USA), a permeabilization step enabled the access for Alexa fluor anti-rabbit 488 (1:500, Thermo Fisher, Rockford, IL, USA) to both extra- and intracellular bacteria. Consequently, extracellular located bacteria are visualized in yellow, intracellular located bacteria are visualized in green. The last staining step additionally includes Phalloidin 660 (1:250, Thermo Fisher, Rockford, IL, USA) and 4′-6-diamidino-2-phenylindole dihydrochloride (DAPI, 1:50.000) (Calbiochem, Darmstadt, Germany) to stain the actin cytoskeleton and the nucleus, respectively. Inclusion of an additional fixation step after applying the Alexa fluor anti-rabbit 594 antibody to the original protocol resulted in improved staining of the extracellular bacteria.

Invasion analysis and calculation of invasion rates were done as previously described [56]. Briefly, the calculation was carried out with a ZEISS observer Z1 fluorescence microscope with a 63×/1.4 objective lens using Zen software (Carl Zeiss, Jena, Germany). Intra- and extracellular bacteria were counted per defined field; 20 fields per filter were counted and the number of bacteria was extrapolated to the whole filter. Numbers of intracellular and extracellular bacteria were set in relation to the growth of the individual strain determined in parallel (Appendix A depicts a summary of growth curves generated for the presented invasion and adhesion date). Assays were performed at least in triplicates and at least three times.

### 4.7. Measurement of Cell Viability 

To evaluate the effect of *H. influenzae* infection on cell viability, a live/dead-Assay (Thermo Fisher Scientific, Darmstadt, Germany) was performed according to the manufacturer’s instructions. Staining was subsequently documented using a fluorescence microscope. Living cells can be identified due to their green staining. Intracellular esterase activity converts the virtually nonfluorescent cell-permeant calcein AM to fluorescent calcein. The second dye, ethidium homodimer, only enters cells with impaired plasma membrane integrity. Binding to DNA enhances fluorescence intensity, resulting in an intensive red staining of dead cells. 

### 4.8. Ultrathin Sections, Transmission and Scanning Electron Microscopy

Upon the course of 6 h of infection, cell culture inserts were fixed with 1% glutaraldehyde in cacodylate buffer. Since EM sections and subsequent microscopy were kindly performed in the lab of Prof. Rohde insert membranes were re-fixed with 2% glutaraldehyde and 5% formaldehyde in HEPES buffer for 1 h. Following fixation cells were contrasted in 1% aqueous osmium tetroxide.

Cells applied to TEM subsequently were dehydrated applying serial steps of ethanol (10%, 30% and 50%). The highest ethanol concentration (70%) additionally contained 2% uranyl acetate and was applied for overnight incubation. Membranes were finally dehydrated up to 100% ethanol and infiltrated stepwise with low viscosity (LV) resin (1:2 parts resin:ethanol, 1:1 parts, 2:1 parts, pure LV resin) applying the hard formula. Polymerization of LV resin was performed at 75 °C and ultrathin sections were cut *en face* using a diamond knife. Incubation with uranyl acetate and lead citrate resulted in counterstaining of ultrathin sections. The sections were than evaluated by a Zeiss TEM910 with calibrated magnifications at an acceleration voltage of 80 kV. A slow-scan camera was used for image recording. Contrast and brightness were adjusted by Adobe Photoshop 11.0.

Cells to be analyzed by SEM were fixed as mentioned above. After washing with TE buffer (10 mM EDTA, 2 mM EDTA, pH 7.0), a graded ethanol (10%, 30%, 50%, 70%, 90% and 100%) dehydration series was performed on ice with 10 min incubation for each step. Membranes were left in 100% ethanol to adapt to room temperature before applying a final change in 100% ethanol at room temperature and subsequent critical-point drying by liquid CO_2_ (CPD 030, Bal-Tec, Liechtenstein). Finally, cell layers were sputter-coated (SCD 500, Bal-Tec, Liechtenstein) with gold/palladium (80/20). Examination was carried out in a field emission scanning electron microscope Zeiss Merlin (Oberkochen) using the Everhart Thornley HESE2-detector and the inlens SE-detector in a 25:75 ratio at an acceleration voltage of 5 kV. Zeiss SEMSmart V 5.05 software was used for image recording. Contrast and brightness were adjusted with Adobe Photoshop 11.0.

### 4.9. Statistical Analysis

For statistical analysis SAS, release 9.4 (SAS Institute Inc., Cary, NC, USA) was used. Since not all data could be considered as approximately normally distributed and because of the rather small sample sizes, we preferred non-parametric statistical tests. Kruskal–Wallis test was used to compare multiple strains. In the case of a significant test result, global testing was followed by Mann–Whitney U tests for pairwise comparisons to identify differences between groups. Due to relatively low sample sizes, exact *p*-values were calculated. To avoid multiple testing error, we used Bonferroni-correction. All data are shown as mean and median together with quartiles, minimum and maximum.

## Figures and Tables

**Figure 1 ijms-21-05739-f001:**
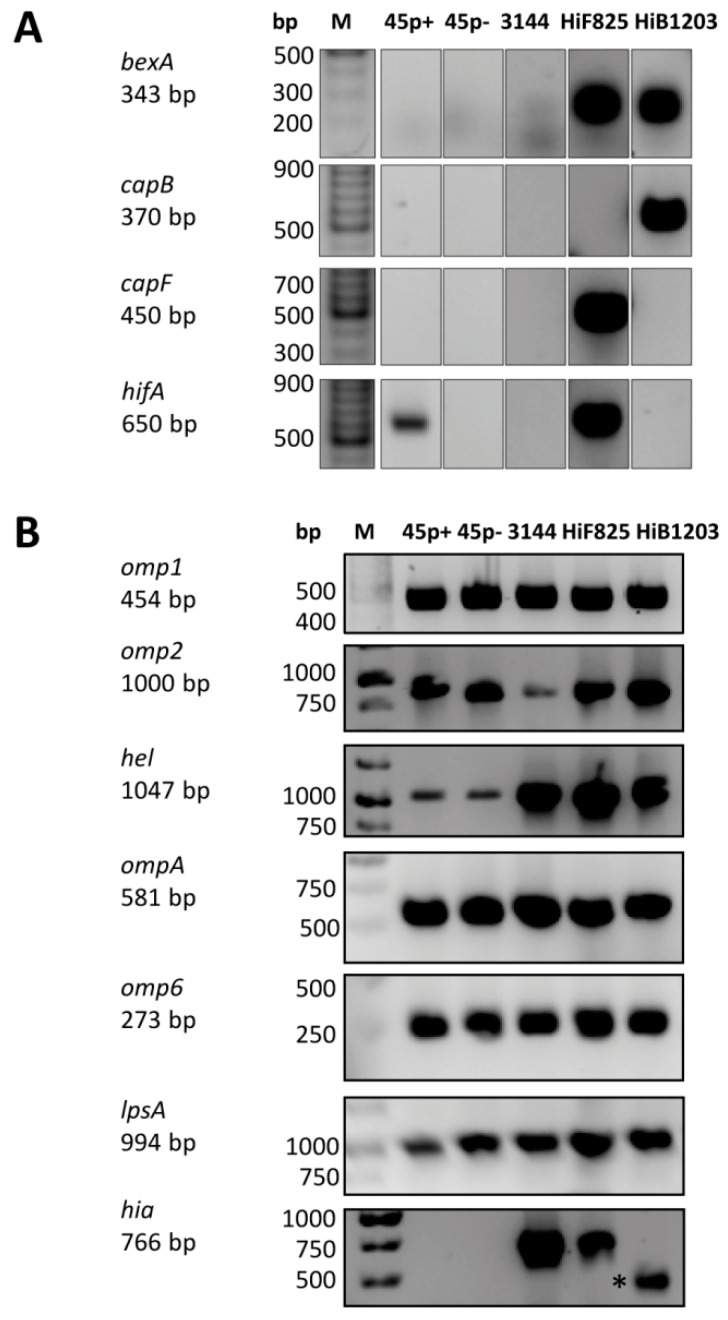
Genomic analyses of the indicated bacterial strains. (**A**) Verification of bacterial strains due to presence of the serotyping capsule gene (*capB* and *capF*), the *bexA* gene responsible for the polysaccharide export to the cell surface and the *hifA* gene enabling isolates to express fimbria. (**B**) Analysis for presence of genes for expression of several virulence factors such as outer membrane proteins (*omp1, omp2, hel, ompA* and *omp6*), adhesin (*hia*) and the glucosyltransferase (*lpsA*). PCR was performed as described in material and methods and the corresponding oligonucleotide primers are shown in Table 1. A truncated amplification product was generated with DNA from HiB1203 and is marked with an asterisk (*). Varying band intensity is due to non-standardized amounts of genomic DNA template. bp, base pairs.

**Figure 2 ijms-21-05739-f002:**
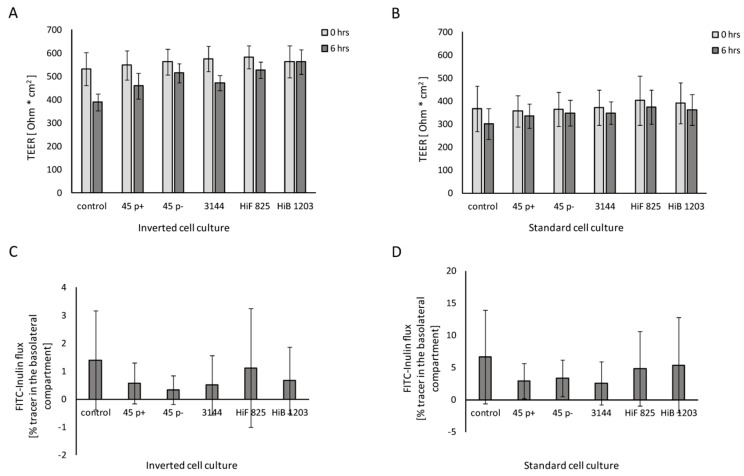
Barrier integrity of HIBCPP cells infected with the indicated NTHI, HiF and HiB isolates Figure 0 h, black bars). TEER values of cells cultured in: (**A**) the inverted cell culture system; and (**B**) the standard cell culture system at the time of infection (time point 0 h, grey bars) and at the end of infection (time point 6 h, black bars). Percentage of FITC-labeled inulin flux of HIBCPP cells grown in: (**C**) the inverted cell culture system; and (**D**) the standard culture system. Infection was carried out for 6 h, and uninfected control cells were analyzed in parallel. The FITC-inulin flux was measured in apical-to-basolateral direction and is expressed as percentage of tracer in the basolateral compartment. Each experiment was repeated at least three times with at least three replicates. All data are shown as mean and error bars depict standard deviation.

**Figure 3 ijms-21-05739-f003:**
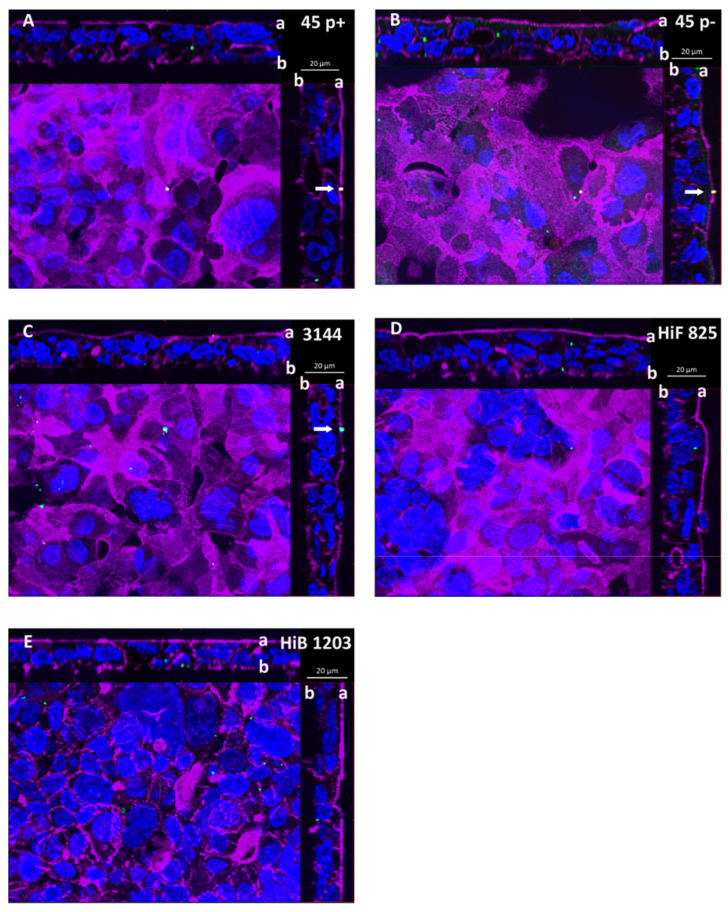
DIF sections of HIBCPP cells depicting cells basolaterally infected with the indicated *Haemophilus* strains. Cells were grown in the inverted cell culture system and challenged with a MOI of 20 of: (**A**) the nasopharyngeal NTHI isolate 45 p+ (fimbriated variant); and (**B**) the nasopharyngeal NTHI isolate 45 p− (non-fimbriated variant); (**C**) the NTHI CSF isolate 3144; and the encapsulated *H. influenzae* CSF isolates (**D**) HiF 825 and (**E**) HiB 1203. After 6 h of infection, cells were subjected to DIF staining to analyze invasion. Intracellular bacteria are visualized in green and extracellular bacteria in yellow. DAPI (blue) and phalloidin (purple) were used to stain the cell nuclei and the actin cytoskeleton, respectively. (**A**–**E**) Apotome images of basolaterally infected HIBCPP cells. The center part of each panel represents the xy *en face* view. An overlay of selected slices through the z-axis intensifies images for presentation. The top and side parts of each panel represent the cross-sections through the z-plane of slices in the overlay. In the cross sections, the apical cell side facing the top and the right side, respectively, is labeled with “a” and the basolateral cell side facing the center part is labeled with “b”. A characteristic distribution for the individual strains is represented. Invaded bacteria (green) appear as single specimen. Extracellular bacteria (yellow) adhering to the apical cell side are marked with a white arrow. These bacteria represent specimen that have transmigrated through the HIBCPP cell layer from the basolateral side still being attached to the apical cell side.

**Figure 4 ijms-21-05739-f004:**
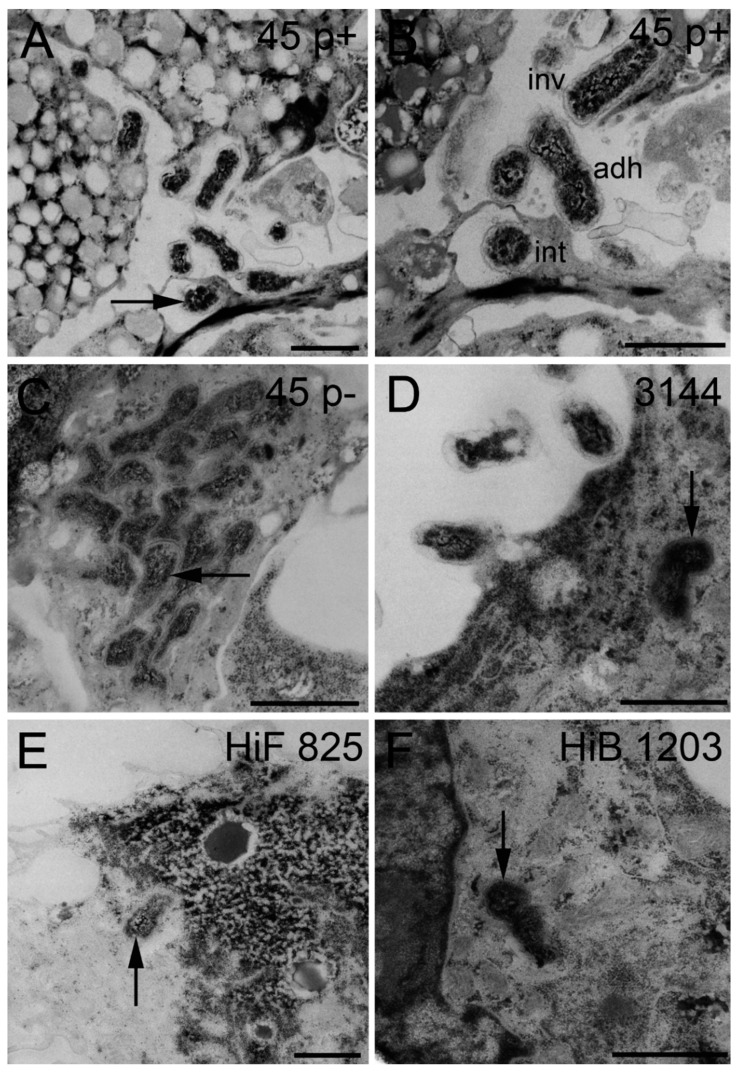
TEM sections of HIBCPP cells grown in the inverted cell culture system infected with indicated NTHI and *H. influenzae* isolates for 6 h. (**A**–**F**) TEM images of intracellular bacteria (black arrow), which invaded from the blood facing side. (**B**) Different phases of the pathogenesis progress—bacteria adhering to HIBCPP cells, invading into HIBCPP cells and already intracellular of a HIBCPP cell. Scale bars represent 1 µm. inv, invaded; adh, adherend; int, intracellular. “p+” means fimbriated and “p−“ means non-fimbriated.

**Figure 5 ijms-21-05739-f005:**
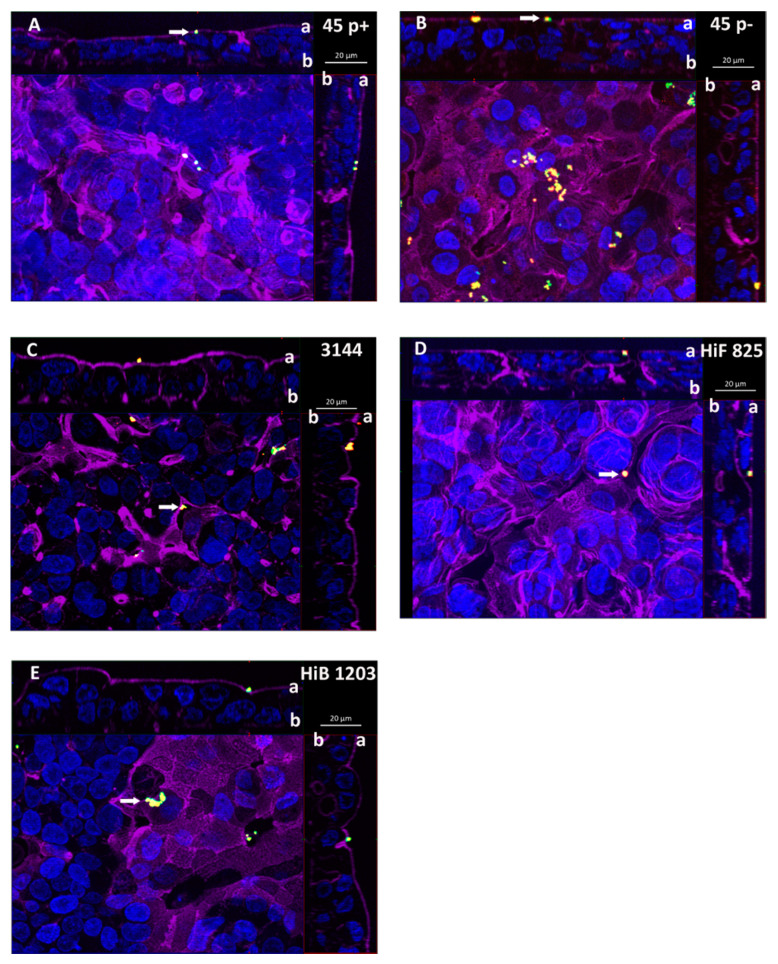
DIF sections of HIBCPP cells apically infected with the indicated *Haemophilus* strains. Cells were grown in the standard cell culture system and challenged with a MOI of 20 of: (**A**) the nasopharyngeal NTHI isolate 45 p+ (fimbriated variant); (**B**) the nasopharyngeal NTHI isolate 45 p− (non-fimbriated variant); (**C**) the NTHI CSF isolate 3144; and with the encapsulated *H. influenzae* CSF isolates (**D**) HiF 825 and (**E**) HiB 1203. After 6 h of infection cells were subjected to DIF staining to analyze invasion. Intracellular bacteria are visualized in green and extracellular bacteria in yellow. DAPI (blue) and phalloidin (purple) were used to stain the cell nuclei and the actin cytoskeleton, respectively. (**A**–**E**) Apotome images of apically infected HIBCPP cells. The center part of each panel represents the xy *en face* view. An overlay of selected slices through the z-axis intensifies images for presentation. The top and side part of each panel represent the cross-sections through the z-plane of slices in the overlay. In the cross sections the apical cell side facing the top and the right side, respectively, is labeled with ”a” and the basolateral cell side facing the center part is labeled with “b”. A characteristic distribution for the individual strains is represented. Almost no invaded bacteria (green) could be detected in the standard cell culture infected cells. Extracellular bacteria (yellow) adhering to the apical cell side (CSF facing side) are marked with a white arrow.

**Figure 6 ijms-21-05739-f006:**
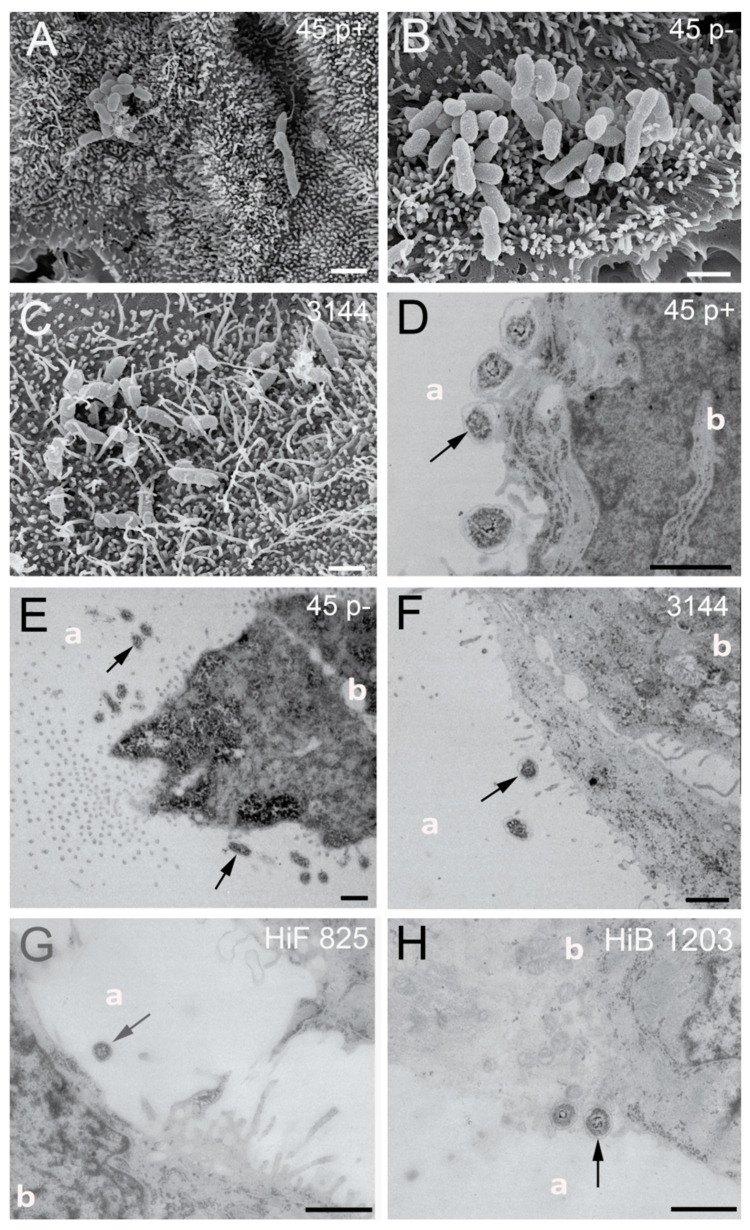
SEM and TEM sections of HIBCPP cells grown in the standard cell culture system infected with indicated NTHI and *H. influenzae* isolates for 6 h: (**A**–**C**) SEM images of NTHI strains 45 p+, 45 p− and 3144, which are located at the microvilli covered apical cell side; and (**D**–**H**) TEM images of bacteria adhering to the cells sitting on or next to microvilli of the HIBCPP (black arrows). Almost no intracellular bacteria could be detected in the standard culture grown HIBCPP cells. The apical and basolateral cell sides are labeled with “a“ and “b”, respectively. Scale bars represent 1 µm.

**Figure 7 ijms-21-05739-f007:**
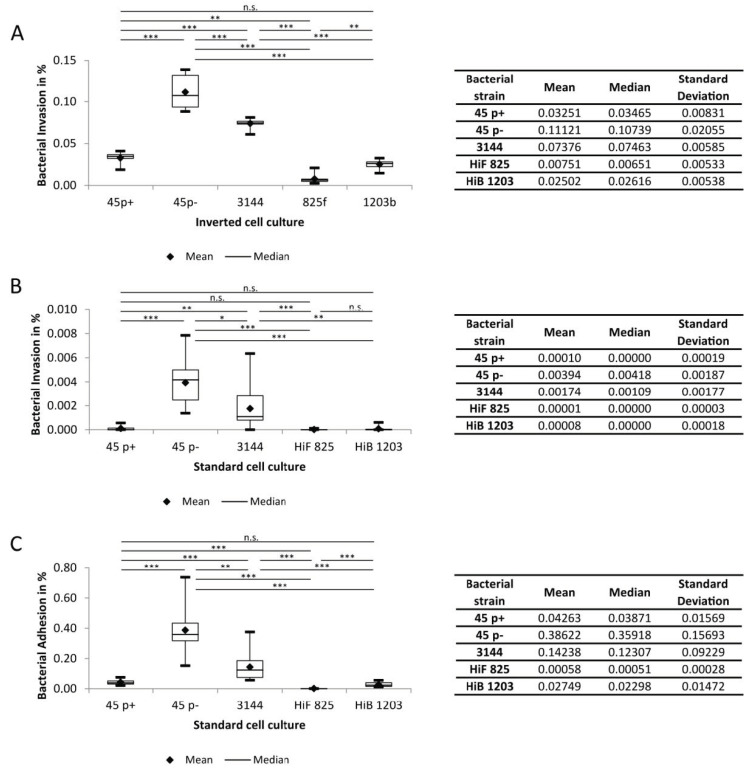
Invasion and Adhesion rates of indicated *H. influenzae* strains. Invasion upon infection with a MOI of 20 was counted after 6 h of infection: (**A**) invasion rates in the inverted cell culture (*n* = 9); (**B**) invasion rates in the standard cell culture ((*n* = 12, *n* = 13 for HiF 825); and (**C**) adhesion rates in the standard cell culture. Each experiment was repeated at least three times with at least three replicates. All data are shown as mean and median together with quartiles, minimum and maximum. The box depicts the interquartile range (25% and 75%). * *p* < 0.05; ** *p* < 0.01; *** *p* < 0.001; n.s., not significant.

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
