# Peer review of "Non-Typeable Haemophilus influenzae Invade Choroid Plexus Epithelial Cells in a Polar Fashion"

_ijms, 2020, doi:10.3390/ijms21165739_

Round 1
Reviewer 1 Report
The manuscript by Wegele et al investigate the invasion of choroid plexus epithelial cells by non-tapeable Haemophilus influenziae. The authors use a series of carefully designed experiments to evaluate the interactions of H. influenziae with the apical and basolateral surface of a polarised choroid plexus epithelial cell line grown on Transwell filters. I have some suggestions for improvement of the manuscript before acceptance:
- The error bars in Figure 2C and D are extremely large. Is this a reflection of the relatively low TEER values observed in these cells? It would be useful to comment on this.
- It would be useful to include a positive control in both the TEER and inulin flux assays, such as TNF-a, shown in Figure 2.
- While the authors convincingly show that bacteria invade cells from the basolateral (blood side) of the cells, the authors do not seem to have demonstrated that bacteria can completely cross the cells and be detected on the brain side of the cells. Have the authors checked the culture medium on the apical side of the cells following basolateral inoculation to verify that bacteria can completely transmigrate across these cells?
Reviewer 2 Report
The authors investigate the invading mechanism(s) for NTHI and found polar invasion in HIBCPP cells. The authors used imaging-based system to address how NTHI invade into cells. However, an alternative(more straightforwardly quantitative) method would be needed to double confirm the evidence visualized by Confocal and TEM/SEM.
Figure 2. Lack of Positive control to show what decreased TEER and inulin leakage looks like.
Figure 3 and 5. Why are there no adhered bacterial cells?
Figure 7. A concern on image-based invasion and adherence quantification: A more accurately quantifiable assay, such as gentamycin killing assay, will be needed besides confocal data. Confocal data can tell distribution, but it is hard to tell the total number.
Figure 3,4,5,6. Images of no HI cells are needed to compare for cell integrity.
Figure 3,4,5,6. Label apical/basal side for clarification.
Line 116. "...capsule marjers" --> capsule markers.
Line 139-142. The hemagglutination data should either be shown or not mentioned.
Line 180-182. It is confusing that figure 2C and D showed no significant difference in comparison but a very low and higher inulin influx were mentioned in the context.
Line 279-280. Delete the sentence. The sentence describes the result therefore not fitting into figure legend.
Line 511. A growth curve should be put in within the paper somewhere to fit in to this sentence " ...bacteria were set in relation to the growth of the individual strain determined in parallel..."
Line 513-518. Where in the context/figure did the authors use this live/dead staining?
Round 2
Reviewer 2 Report
The authors have addressed the questions appropriately with either evidence and/or references. The paper is ready to go.
Figure S2 - Can author give significance between TEERS for better clarification.
Author Response
Dear Reviewer,
we performed statistical analysis of the data represented in figure S2 and added the following information to the figure legend:
"0 hrs TEER: Kruskal-Wallis test reveals p=0,9327, thus there is no difference between different treatments. 2,5 hrs TEER: Kruskal-Wallis test reveals p=0,0156, thus there is a significant difference between the treatments. Inulin: Kruskal-Wallis Test reveals p=0,0188, thus there is a significant difference between the treatments. In the two latter data sets a subsequent Wilcoxon 2 sample test is not possible due to the low sample size. The greatest differences in rank sums and medians, however, were observed between the untreated control and the highest Cytochalasin concentration of 1 µg/ml followed by 0,2 µg/ml and 0,1 µg/ml, which due to sample size, however, are not detectable."